# Direct Synthesis of MOF-74 Materials on Carbon Fiber Electrodes for Structural Supercapacitors

**DOI:** 10.3390/nano14020227

**Published:** 2024-01-20

**Authors:** David Martinez-Diaz, Pedro Leo, David Martín Crespo, María Sánchez, Alejandro Ureña

**Affiliations:** 1Materials Science and Engineering Area, Escuela Superior de Ciencias Experimentales, Universidad Rey Juan Carlos, C/Tulipán s/n, 28933 Móstoles, Spain; d.martincre@alumnos.urjc.es (D.M.C.); alejandro.urena@urjc.es (A.U.); 2Departament of Chemical and Enviromental Technology, Universidad Rey Juan Carlos, C/Tulipán s/n, 28933 Móstoles, Spain; pedro.leo@urjc.es; 3Instituto de Tecnologías Para la Sostenibilidad, Universidad Rey Juan Carlos, C/Tulipán s/n, 28933 Móstoles, Spain

**Keywords:** metal–organic framework, MOF-74 materials, carbon fiber, energy storage, structural supercapacitor

## Abstract

The use of fossil fuels has contributed significantly to environmental pollution and climate change. For this reason, the development of alternative energy storage devices is key to solving some of these problems. The development of lightweight structures can significantly reduce the devices’ weight, thereby reducing energy consumption and emissions. Combining lightweight structures with alternative energy storage technologies can further improve efficiency and performance, leading to a cleaner and more sustainable system. In this work, for the first time, MOF-74 materials with different divalent metal ions have been synthesized directly on carbon fiber, one of the most widely used materials for the preparation of electrodes for supercapacitors with structural properties. Different techniques, such as nitrogen adsorption–desorption isotherms, cyclic voltammetry or galvanostatic charge–discharge, among others, were used to evaluate the influence of the metal cation on the electrochemical capacitance behavior of the modified electrodes. The Co-MOF-74 material was selected as the best modification of the carbon fibers for their use as electrodes for the fabrication of structural supercapacitors. The good electrochemical performance shown after the incorporation of MOF materials on carbon fibers provides a viable method for the development of carbon fiber electrodes, opening a great variety of alternatives.

## 1. Introduction

Today, a growing concern for global sustainability and energy demand can be observed. The use of fossil fuels as the main source of energy has contributed significantly to environmental pollution and climate change. The combustion of these fuels releases harmful emissions such as carbon dioxide, nitrogen oxides, and particulate matter into the atmosphere, which have adverse effects on human health and the environment. The transportation sector is one of the major contributors to this problem, with vehicles emitting large amounts of greenhouse gases and air pollutants [1]. For this reason, the transportation sector must look for alternatives to the use of fossil fuels as fuel for vehicles. Currently, there are two alternatives: the use of electric batteries and the use of hydrogen as an energy source. The development of hydrogen-powered vehicles is not yet a reality, but the idea is still being researched [2,3]. Battery electric vehicles (BEVs), on the other hand, are a current alternative and are attracting industry interest, as they significantly reduce urban air pollution. But, undoubtedly, one of the factors that most affects these vehicles is the duration of their batteries, since their autonomy is low compared to hybrid and conventional vehicles [4]. Therefore, it is necessary to develop new energy sources or storage devices to make them economically profitable [5,6,7,8].

To address these issues, batteries and supercapacitors (SCs) are the two primary technologies used in energy storage systems [9,10]. While batteries tend to have higher energy densities, they also have lower power densities compared to SCs [11]. Both technologies offer advantages over traditional systems in terms of efficiency, performance, and environmental impact. Supercapacitors, also known as electrochemical capacitors, have gained significant attention [12] due to their ability to deliver high power densities and a long cycle life, as well as their balance of energy and rate performance [10,13,14]. Supercapacitors can be classified into two main groups, considering the active materials used and the charge storage mechanism [14]. One is based on surface redox reactions, known as pseudo-capacitors. Conductive polymers or transition metal oxides are examples of pseudo-capacitor-active materials [15]. Another one is based on the use of materials that can store electrical energy on their internal surface through the use of electrostatic forces, known as electrochemical double-layer capacitors (EDLC) [6,16]. Typically, this type of SC uses carbon-based materials [17] such as microporous or mesoporous carbons [18,19,20], graphene [21,22,23], or carbon nanotubes [22,24,25] as the active material.

As an alternative for this last supercapacitor type, metal organic frames (MOFs) have shown potential for their use as electrode materials in SCs [26] due to their large surface area, structural controllability, exceptionally high crystallinity, and active centers [14]. This allows them to meet the most important requirements for the use of any material in a SC to achieve high performance in terms of power density and energy storage capacity, such as high conductivity and large surface areas accessible to electrolytes [27]. MOFs can be used for the preparation of supercapacitors in various ways, such as using the MOFs without modification, destroying the MOFs to obtain metal oxides, or pyrolyzing the MOFs to achieve nanoporous carbons [28]. Normally, all these materials (carbons, graphene, carbon nanotubes, MOFs) need to be supported onto another material to be able to obtain a functional electrode for the preparation of a supercapacitor [6]. Carbon-based materials were widely used for this purpose in energy storage devices [8,28]. More specifically, carbon fiber fabric (CFF) is one of the most commonly used due to its high electrical conductivity, easy fabrication process, and low cost. Additionally, the use of carbon fiber (CF) opens the possibility of applying these electrodes for structural purposes. The use of CF as a structural reinforcement has been extensively studied in multiple sectors such as automotive, aerospace, and construction [29] due to its excellent mechanical properties for the fabrication of lightweight structures [30]. Additionally, lightweight materials, such as CFF, can significantly reduce the final weight of the structure, leading to lower energy consumption and emissions. Moreover, there is currently an increasing effort in the development of multifunctional materials based on carbon fiber reinforced polymers (CFRPs) that allow the combination of the good mechanical performance of the carbon fiber with other interesting properties [21,22,31]. In this regard, CFRP-based supercapacitors can combine structural and energy storage capabilities in one device [32]. Combining lightweight structures with supercapacitors can further improve the efficiency and performance of vehicles, leading to a cleaner and more sustainable transportation system. Overall, the adoption of alternative energy storage technologies and lightweight structures in the transportation sector can significantly reduce the environmental impact of the sector, contributing to a cleaner and more sustainable future.

In this work, in order to increase the surface area of CFF electrodes, MOF-74 materials with different divalent metal ions (M = Mn^2+^, Ni^2+^, Co^2+^, Mg^2+^, Cu^2+^ and Zn^2+^) were directly synthesized on the fabric for the later fabrication of structural supercapacitors by vacuum assisted resin infusion molding (VARIM). It is worth noting that the achieved structural properties were a result of utilizing carbon fiber fabric and an epoxy-based resin, which are commonly employed as reinforcement and matrix materials, respectively, in structural applications. This well-known MOF-74 family constituted by 2,5-dihidroxyterephtalic acid as an organic ligand was selected because of its high specific surface area (approximately 1000 m^2^/g) [33], the present infinite rod units containing a high density of open metal sites, its confirmed electrical conductivity [34], and its ability to be synthesized under conditions that do not degrade the carbon fiber fabrics [33,35,36,37]. Additionally, due to the isostructural properties of the MOF-74 family (Figure 1) [38], different active metals were used to ascertain the influence of the metal cation on the electrochemical capacitance behavior of the electrodes.

## 2. Materials and Methods

### 2.1. Preparation of Electrodes

In this study, a series of metal–organic framework (MOF) materials were synthesized on the surface of carbon fiber fabrics (CFF) to produce electrodes for structural supercapacitors. The CFF used was a five-harness satin weave (AS4C 3k 5H), supplied by Hexcel. Prior to the synthesis, the polymer sizing was removed by cleaning the fabric in acetone and submerging it in nitric acid for 1 h to remove the original fiber coating. After the sizing removal, the CFF was cut into small samples of 4.0 × 4.0 cm. The synthesis procedure of the M-MOF-74 materials (M = Mn, Ni, Co, Mg, Cu, and Zn) was outlined in Table 1. Starting materials were purchased from Sigma-Aldrich or TCI and used without further purification. In all cases, the synthesis of the different MOF-74 materials was carried out following the same procedure. In a beaker, a joint solution of the corresponding metal source and 2,5-dihydroxyterephthalic acid was prepared in a previously prepared mixture of solvents. Once these reagents had been dissolved, they were transferred to an ISO bottle, and the CFF was also introduced. The ISO bottle was capped tightly and placed in an oven at the desired temperature during the reported time (see Table 1).

### 2.2. Preparation of the Supercapacitors

After the electrode preparation, the modified CFFs were used as electrodes for the preparation of structural supercapacitors by means of a vacuum-assisted resin infusion molding (VARIM) process. In this process, two MOF-74 modified CFF electrodes were separated by two glass fiber layers, and then the polymeric matrix was infused by vacuum and cured at 140 °C for 8 h to obtain the final CFRP structure. In this case, the polymeric matrix combines structural and electrolytic properties because of the mixture of two epoxy resin systems with an ionic liquid. One of the epoxy systems is based on the use of the commercial monomer for structural applications, Araldite LY556, with Araldite XB3473 as a hardener, in a weight ratio of 100:23, respectively. The other epoxy system was the combination of the monomer poly(ethylene glycol) diglycidyl ether (PEGDGE) with 4-4′-diaminodiphenylsulfone (DDS), in a weight ratio of 100:35, respectively. Moreover, to achieve the electrolytic properties on the matrix, 1-ethyl-3-methyl-imidazolim bis-(trifluoromethylsulfonyl)imide (ILE) was used as an ionic liquid. Finally, alumina nanoparticles (Al_2_O_3_), with a maximum size of 13 nm and 99.8% purity, were used as a mechanical reinforcement of the matrix. To summarise, the composition of the matrix was 44.2 wt% of the LY556/XB3473 system, 23.8 wt% of the PEGDGE/DDS system, 30 wt% of ILE, and 2 wt% of Al_2_O_3_ nanoparticles. More details about the matrix optimization and the preparation procedure were reported previously by Del Bosque et al. [39]. It must be pointed out that when using the word structural supercapacitor, the structural term is associated with the use of a reinforcement and an epoxy-based matrix system for composite manufacturing, both of which are normally used for structural purposes.

### 2.3. Characterization

X-ray powder diffraction (XRD) patterns were acquired on a Philips X’pert diffractometer (Eindhoven, The Netherlands) using Cu K*α* radiation. The data were recorded from 5 to 50° (2*θ*) with a resolution of 0.01°. Nitrogen adsorption–desorption isotherms at 77 K were measured using AutoSorb equipment (Quantachrome Instruments, Boynton Beach, FL, USA) in order to analyze the specific surface area of the synthesized materials. Fabric surface characterization and MOF morphology were analyzed using images taken from scanning electron microscopy (SEM, S-3400 N from Hitachi, Tokyo, Japan).

Cyclic voltammetry (CV) tests on modified CFF were performed using a Metrohm Autolab PGSTAT302N potentiostat (Herisau, Switzerland). This experiment was carried out at room temperature with a three-electrode cell using an Ag/AgCl reference electrode in a 2 M potassium chloride (KCl) ethanol electrolyte. It must be pointed out that an aqueous electrolyte cannot be used due to the observed degradation of MOF-74 in water [40].

The working electrode consisted of a single wire partly immersed in the electrolyte and electrically contacted at the dry end. Tows were weighted and measured before the test, and the active mass (*m_a_*) of the working electrode was determined from the immersed length. Different scan rates (5, 10, 25, 50, 75, and 100 mV/s) using a potential window (Δ*V*) from −0.1 V to 0.1 V were tested on a representative specimen from each condition. Samples were subjected to 10 consecutive voltammetry cycles to determine the specific capacitance. Specific capacitances (*C_sp_*) were calculated from the CV curves of three-electrode system with Equation (1).
(1)$$C_{sp}=\frac{\int_{V_{o}}^{V_{f}}IdV}{s\cdot \Delta \cdot m_{a}}$$

Cyclic voltammetry (CV) and galvanostatic charge–discharge (GCD) tests have also been carried out in structural supercapacitor composites. In this case, working and counter electrodes are connected to each MOF-modified CFF electrode. Electrochemical response was recorded in a voltage window of 0 to 1 V at different voltage scan rates (from 5 to 100 mV/s). Galvanostatic charge–discharge tests were performed at a current density of 1 A/g from 0 to 1 V for 5 cycles.

## 3. Results and Discussion

### 3.1. Characterization of MOF-74 Electrodes

Prior to the characterization of the synthesized MOF-74 materials, it is necessary to confirm the direct synthesis of these materials on the CFF, since it is a key step towards the goal of simplifying the fabrication of structural supercapacitors. As can be seen in Appendix A, in all the syntheses performed, a complete and homogeneous deposition is observed over the entire surface of the CFF to be coated. The treatment performed prior to the CFF is indispensable to improving the coating of the CFF surface with MOF, since when it is not performed, the degree of deposition achieved is lower and does not occur homogeneously. All the synthesized materials were characterized by the usual physicochemical techniques for this type of porous materials. The XRD patterns obtained from each sample of M-MOF-74 (M = Mn, Ni, Co, Mg, and Zn) are presented in Figure 2. All the synthesized materials show the main diffraction reflections characteristic of the MOF-74 structure (6.8° and 11.8°), confirming the presence of this phase in the different synthesized materials. It is also worth mentioning that no additional reflections are found that do not coincide with the simulated pattern, thus confirming the presence of a single crystalline phase. In spite of maintaining the same structure, differences in signal intensities are observed. This could be associated with a difference in the sputtering of the samples or with the different degrees of crystallization of the materials [35]. It should be noted that the MOF-74 structure has fully accessible metal centers, a characteristic that is not present in many MOF materials and is key to the development of the MOF-74 structure [41].

On the other hand, it is necessary to know the textural properties of the synthesized materials since this parameter may have an impact on an increase in the final capacity of a supposed supercapacitor manufactured later, as reported by other authors [42]. The nitrogen adsorption and desorption isotherms performed are shown in Appendix A, where it can be observed that all adsorption–desorption isotherms are basically type I, revealing a permanent microporosity. The BET surface area and pore volume are collected in Table 2, noting that the highest surface area achieved is at 1127 m^2^/g for the Co-MOF-74 sample, although similar values were also obtained for the Cu-MOF-74 and Ni-MOF-74 samples. However, for the remaining three samples, lower surface area values were obtained compared to the other three samples. Although all the materials have the same structure, the atom size of each of the active metals used is different, leaving a different pore size in each case [43]. Regardless, of the metal composing the MOF-74 structure, the direct synthesis of MOFs on carbon fiber fabric improves the textural properties considerably, as the CFF shows a specific surface area of 18 m^2^/g [44].

Although the MOF-74 family of materials is isostructural, the morphology of the crystalline particles obtained from each of the materials differs from each other, as can be seen in the SEM images summarized in Figure 3. This fact is tentatively associated with the solubility of the metal source, which determines the magnitude of the M-MOF-74 crystal size [45]. Figure 3a shows the capture taken for the carbon fiber strand on which the Mn-MOF-74 material was deposited. Crystals of large size and thickness are observed on the fiber itself, in fact, most of them are thicker than the carbon fiber itself. The images taken for the Ni-MOF-74 wick correspond to Figure 3b, and it can be clearly observed that the material formed has a much smaller size than the Mn-MOF-74 material; moreover, a much smaller amount of Ni-MOF-74 has been formed compared to the previous sample. For the Co-MOF-74 sample (Figure 3c), crystals of an intermediate size formed between the two materials previously. As for the distribution of the crystals along the wick, it can be seen how they are distributed in the form of colonies. In the case of the Cu, Zn, and Mg MOF-74 materials, they present a smaller crystal size than those obtained for Mn and Co MOF-74 materials but are larger than Ni according to the scale of the images. Regardless of the MOF-74 material synthesized, crystal growth from the fiber is clearly visible in all images. Despite the observed variation in crystal sizes in the synthesized samples, it is expected that this will not significantly affect the electrochemical properties of the final electrodes. This is because the capacitance was calculated using Equation (1), which takes into account the mass of the material itself.

### 3.2. Electrochemical Properties of MOF-74 Electrodes

In this section, the electrochemical properties of the synthesized electrodes were characterized by cyclic voltammetry. The obtained CV results for the different tested CFFs modified with MOF-74 are presented in Figure 4, where no significant distortions in the curve shapes were observed for any material. In all cases, the faster scan rate, the higher current for the analyzed potential window. First, it is important to highlight that the maximum areas of the CV loop for any MOF coated CF are much larger than the raw CF, indicating that the presence of MOF-74 produces a positive effect on the electrochemical capacitive properties. Moreover, since redox reactions were not involved in the process, the obtained curves did not present the typical duck shape observed with the oxidation or reduction of a metal complex [46]. According to CV measurements, specific capacitances in the range of 0.45 and 1.95 F/g were obtained at 5 mV/s for MOF-74-coated CF in 2.0 M KCL ethanol electrolyte. It is important to point out that, despite the lower conductivity of the used electrolyte, a non-aqueous electrolyte was used due to the degradation problems observed for MOF-74 in water, previously reported elsewhere [40]. The best results were achieved when using cobalt as an active metal, closely followed by nickel and manganese. On the contrary, lower specific capacitances were obtained with zinc, magnesium, or copper. The specific capacitance values obtained with each active metal for the different tested conditions are represented in Figure 4g. These results were also summarized in Appendix A.

The use of cobalt in supercapacitors as a promising active material was also reported by other authors [47], but in a general way using oxides such as Co_3_O_4_ [48,49,50,51,52]. In this work, the good performance of cobalt was also confirmed in the case of using this metal in a MOF structure, better than what was observed in the case of using the other possible metals such as Mn, Ni, Mg, Cu, and Zn. This best result obtained by using Co is directly associated with the best combination of properties in terms of BET surface area and pore volume (Table 2). This allows for increased contact between the active material and the electrolyte, so more electrical energy can be stored on their internal surface through the use of electrostatic forces, resulting in a higher specific capacitance. Currently, it is considered important to find new materials that exhibit high specific capacitance, which can replace the use of representative transition group metal materials like RuO_2_, for which there is a shortage and generate significant environmental toxicity [47,48]. As conclusion for future works after analyzing the obtained results, together with those previously reported by other authors, the synthesis of Co-MOF-74 and its subsequent calcination can be considered an attractive alternative to obtaining a cobalt oxide with a high specific surface area. In this way, the advantageous properties reported for cobalt oxide usage would be combined with the achievement of a structure with a high specific surface area, facilitated by the use of the Co-MOF-74 structure as a template. As a consequence of the good capacitive behavior observed in case of using cobalt as an active metal, the Co-MOF-74 material was selected as the best CFF modification for the fabrication of electrodes for structural supercapacitors.

### 3.3. Electrochemical Properties of Structural Supercapacitors

As previously detailed, a structural supercapacitor was fabricated using two layers of carbon fiber fabrics with Co-MOF-74 as electrodes. Two layers of glass fiber fabric were used to separate both electrodes, and the final structure was obtained by the infusion of a polymeric resin mixture that contained the ionic liquid. The manufactured composite structure presented dimensions of 5 × 5 × 0.4 cm (length, width, and thickness). The structural term has been employed because the used carbon fiber (Hexcel AS4C 3k 5H) and the epoxy systems (Ly556/XB3473 and PEDGE/DDS) are widely used in the manufacture of CFRP composites for structural purposes. Specifically, the assessment of the mechanical properties of these materials, reinforcement, and matrix, was previously analyzed and reported elsewhere [21]. Therefore, as a consequence of this, and since no damage has been observed on the CF in the SEM analysis after the MOF deposition process (Figure 3), it can be concluded that the CFRP manufactured through VARIM can also maintain its structural properties. Figure 5a shows the obtained CV curves for the supercapacitor after connecting a copper sheet to each electrode. In a similar way to the previous CV curves, the highest current was achieved at the highest scan rate. Due to the observed curve shapes, the presence of redox reactions between Co-MOF-74 and the polymeric matrix was not observed. Moreover, the observed capacitive behavior is similar to that of an electric double layer capacitor (EDLC) [6]. These results demonstrate the good anchorage of the Co-MOF-74 material on the carbon fiber, due to their direct synthesis on the fiber surface since it has not been dragged by the resin flow during the VARIM process. The calculated specific capacitances for each analyzed scan rate are represented in Figure 5b. Specific capacitances from 0.18 to 6.21 mF/g were achieved, getting the highest values at the lower scan rates. Figure 5c shows the galvanostatic charge–discharge test carried out with a current density of 1 A/g, on the Co-MOF-74 structural supercapacitor. After the first charge cycle, deviations below 10% on the charge time were achieved for the other cycles. Charging time slowly decreased with each cycle until it stabilized at approximately 240 s. On the contrary, the discharge time has been stable at approximately 177 s since the first cycle, with deviations below 1%.

As a proof of concept, the Co-MOF-74 structural supercapacitor was connected to an LED with an electrical arrangement with the aim of illuminating it. First, the SC was charged at 5.0 V for different times (from 1 to 30 min) and then disconnected from the battery and connected to the LED (Appendix A). As can be seen in Table 3, the lighting time of the LED was higher than the charging time of the structural supercapacitor in all the analyzed cases. The lighting time was 400% higher than the charging time for the shortest time (1 min), 220% higher for the 5-min charge, and 127% higher for the two longest charge times (15 and 30 min).

## 4. Conclusions

For the first time, the direct synthesis of MOF-74 materials on a woven carbon fiber fabric for their use in structural supercapacitors was achieved. Different analytical techniques, such as DRX or BET, confirmed the incorporation of these M-MOF-74 materials (M = Mn, Ni, Co, Mg, Cu, and Zn) with a high degree of dispersion on the carbon fiber fabrics. Moreover, the synthesized materials exhibit specific capacitances in the range of 0.45 and 1.95 F/g at a sweep rate of 5 mV/s, substantially improving the performance of raw carbon fiber. In this way, the high influence of the metal ion has been analyzed and verified due to the great difference in behavior observed despite the fact that the same MOF-74 structure has always been synthesized and similar specific areas have been obtained. In particular, the Co-MOF-74-modified CFF electrodes provide the best results in terms of specific capacitance, due to this material exhibiting the best combination of properties in terms of specific surface area and pore volume. Therefore, Co-MOF-74 was the selected material for the fabrication of a structural supercapacitor by VARIM. The good anchorage of the MOF, due to its direct synthesis on the carbon fibers, was demonstrated since the particles have not been dragged by the resin flow during the VARIM process. At the same time, its behavior as a SC was also demonstrated by CV and GCD during several cycles. Additionally, the proof of concept showed in a simple way that the operation of the system achieved lighting times for an LED that were higher than the SC charging times. All these results confirm that the direct synthesis of MOF materials on carbon fiber is an effective technique to obtain functional electrodes for SC and show a new possibility of use for MOF materials

## Figures and Tables

**Figure 1 nanomaterials-14-00227-f001:**
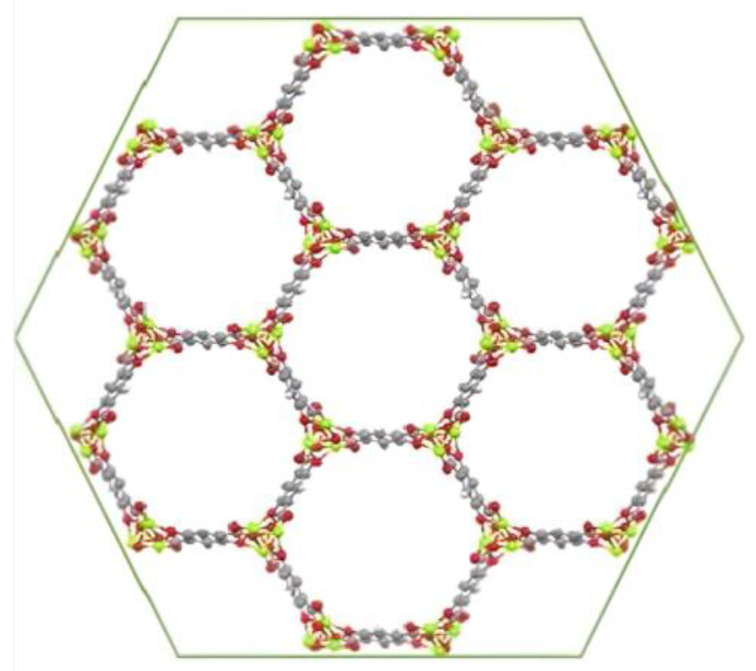
MOF-74 structure [38].

**Figure 2 nanomaterials-14-00227-f002:**
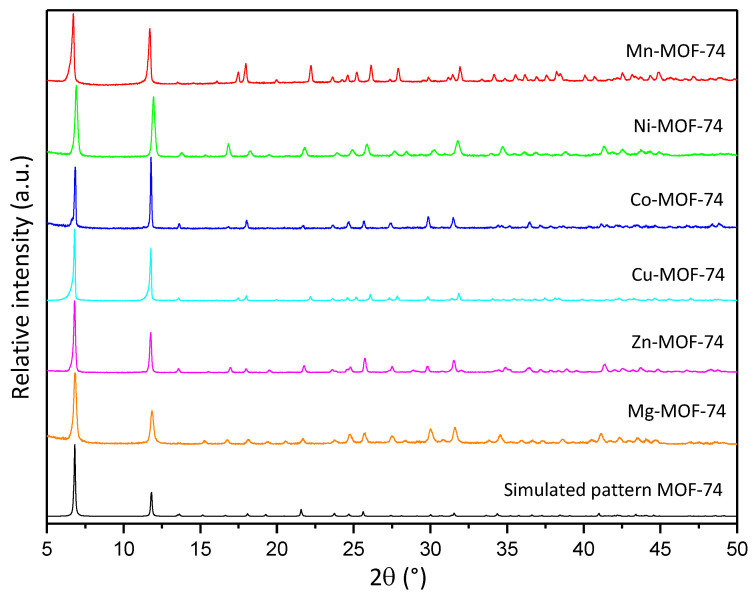
XRD diffractograms of MOF-74 family and simulated pattern.

**Figure 3 nanomaterials-14-00227-f003:**
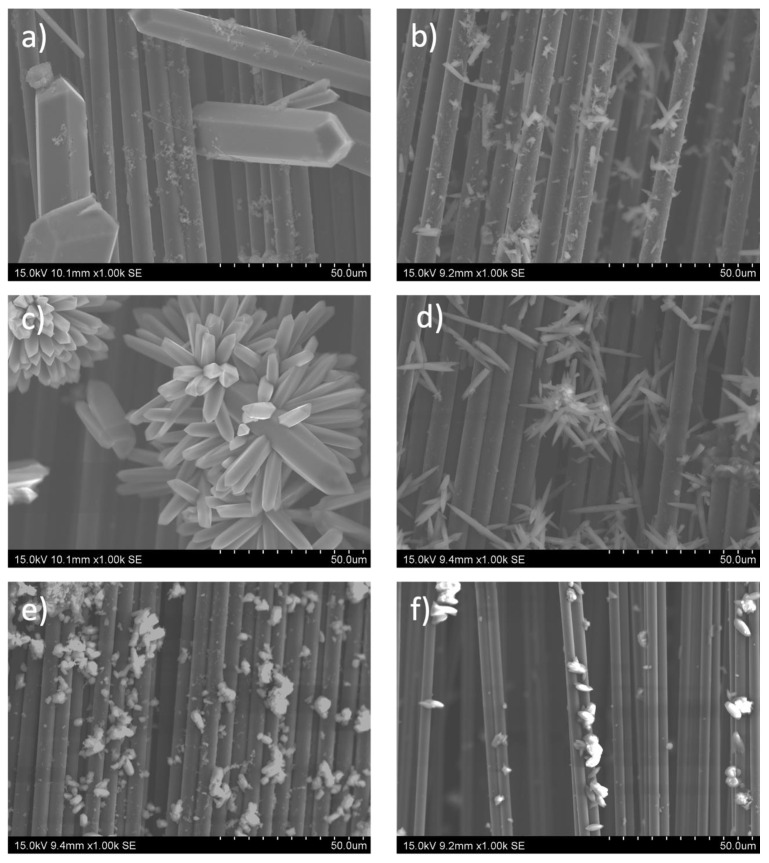
SEM images of carbon fiber fabrics surfaces with different active metals on the M-MOF-74 (M = Mn) (**a**), Ni (**b**), Co (**c**), Cu (**d**), Zn (**e**), and Mg (**f**).

**Figure 4 nanomaterials-14-00227-f004:**
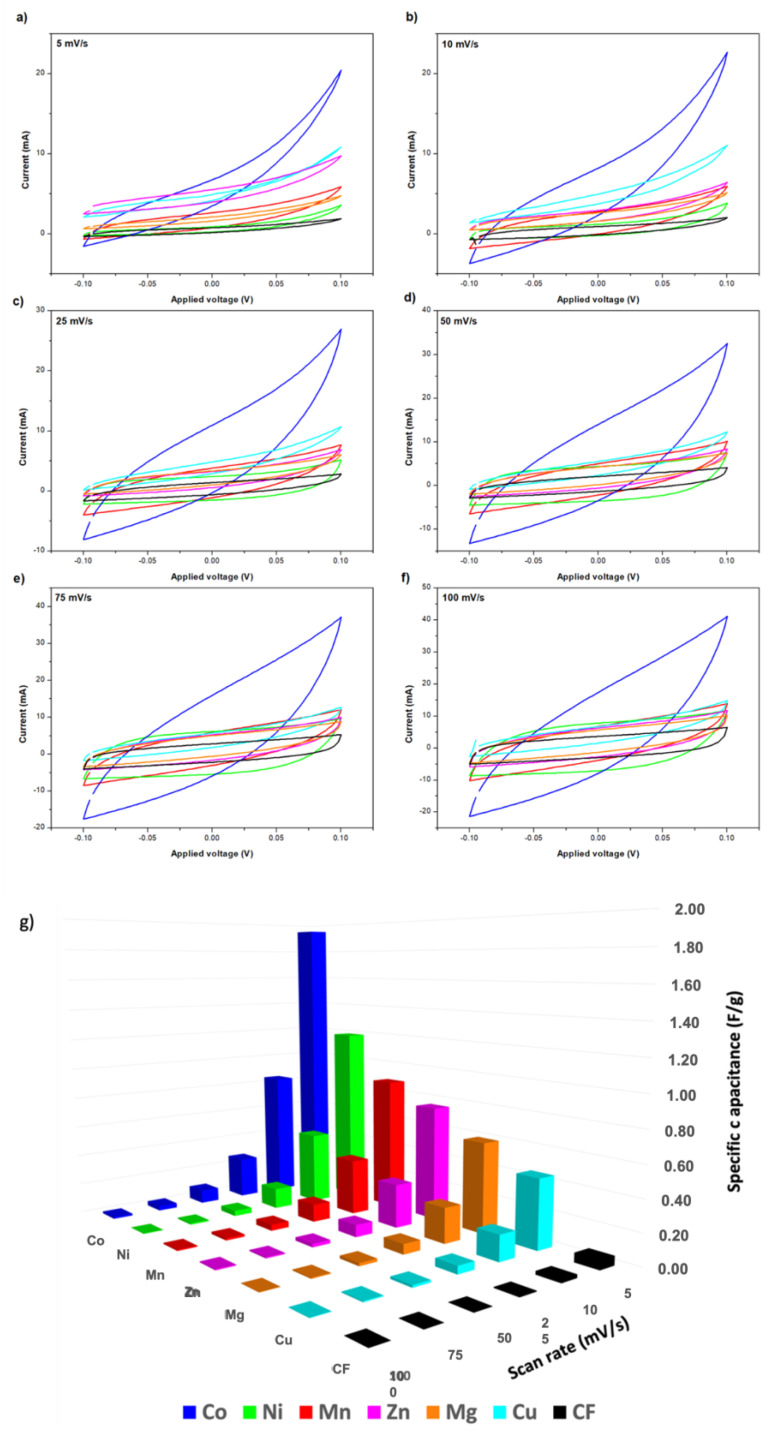
Cyclic voltammetry of electrodes with different active metals on the structure M-MOF-74 at different scan rates (**a**–**f**) and Specific capacitances for the different used active metals on M-MOF-74 structure used as electrodes (**g**). (M = Mn, Ni, Co, Cu, Zn and Mg).

**Figure 5 nanomaterials-14-00227-f005:**
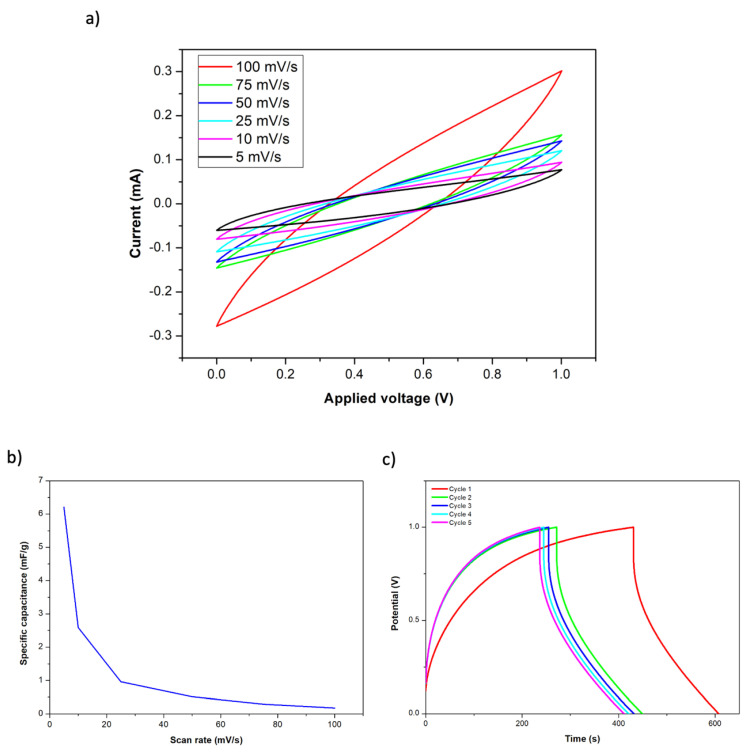
Cyclic voltammetry of Co-MOF-74 supercapacitor at different scan rates (**a**), specific capacitance as a function of current tensity for the Co-MOF-74 supercapacitor (**b**), and galvanostatic charge–discharge curves at 1 A/g (**c**).

**Table 1 nanomaterials-14-00227-t001:** M-MOF-74 synthesis conditions.

Material	Solvent	Metal Source (mmol)	Organic Ligand (mmol)	Temperature (°C)	Time (h)	Ref.
Mn-MOF-74	25 mL 15_DMF_:1_ethanol_:1_H2O_	MnCl_2_·6H_2_O 1.111	0.366	135	24	[35]
Ni-MOF-74	20 mL 1_DMF_:1_ethanol_:1_H2O_	Ni(NO_3_)_2_·6H_2_O 0.818	0.241	100	24	[36]
Co-MOF-74	20 mL 1_DMF_:1_ethanol_:1_H2O_	Co(NO_3_)_2_·6H_2_O 0.867	0.243	100	24	[36]
Cu-MOF-74	25 mL 20_DMF_:1_isopropanol_	Cu(NO_3_)_2_·3H_2_O 1.119	1.119	80	18	[33]
Zn-MOF-74	25 mL 20_DMF_:1_H2O_	Zn(NO_3_)_2_·4H_2_O 0.729	0.243	100	20	[37]
Mg-MOF-74	25 mL 15_DMF_:1_ethanol_:1_H2O_	Mg(NO_3_)_2_·6H_2_O 0.926	0.280	125	20	[36]

**Table 2 nanomaterials-14-00227-t002:** Textural properties of MOF-74 family.

Material	S_BET_ (m^2^/g)	V_p_ (cm^3^/g)
Mn-MOF-74	867	0.43
Ni-MOF-74	1121	0.49
Co-MOF-74	1127	0.54
Cu-MOF-74	1126	0.55
Zn-MOF-74	995	0.44
Mg-MOF-74	710	0.43

**Table 3 nanomaterials-14-00227-t003:** Co-MOF-74 structural supercapacitor charging and LED lighting times.

Supercapacitor Charging Time (min)	LED Lighting Time (min)
1	4
5	11
15	19
30	38

## Data Availability

Data are contained within the article and Appendix A.

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
