# Peer review of "Direct Synthesis of MOF-74 Materials on Carbon Fiber Electrodes for Structural Supercapacitors"

_nanomaterials, 2024, doi:10.3390/nano14020227_

Round 1
Reviewer 1 Report
Comments and Suggestions for Authors
In this work, the authors report the synthesis of Metal-Organic Frameworks, with different metallic cations, on carbon fiber fabrics for the fabrication of electrodes for structural supercapacitors. While this is an important and interesting topic, in my opinion, the manuscript lacks the quality required for publication in Nanomaterials. The physiochemical and electrochemical characterization are deficient and the results are poor and insufficiently substantiated.
In fact, as the main conclusion, the authors state that Co-MOF yields the best results - although this is only based on the specific capacitance values - but do not provide any explanation for this observation. Additionally, no chemical analysis is presented to confirm the presence of the different metallic cations in the structures.
Furthermore, SEM analysis suggests that only small clusters of MOF are deposited on carbon fibers, despite the various colors observed in Figure S1. If, indeed, this is the case, the electrochemical response of the coated materials would be predominantly influenced by the substrate, resulting in minor differences being observed.
In all MOF samples, the voltammograms exhibit a significant tilt concerning the XX axis, indicating a very high resistivity. This is further evident in the charge-discharge curves, which show a pronounced IR drop. The high materials resistivity suggests poor long-term performance of the materials although a comprehensive analysis in this regard was not conducted.
Finally, despite the authors' claim to be developing electrodes for structural supercapacitors, there is no supporting evidence, given the absence of mechanical testing or details regarding adhesion.
For these reasons, in its current form, I believe the manuscript does not meet the requirements for publication.
Comments on the Quality of English LanguageThe English is fine - minor adjustments are required
Author Response
Comments and Suggestions for Authors:
In this work, the authors report the synthesis of Metal-Organic Frameworks, with different metallic cations, on carbon fiber fabrics for the fabrication of electrodes for structural supercapacitors. While this is an important and interesting topic, in my opinion, the manuscript lacks the quality required for publication in Nanomaterials. The physiochemical and electrochemical characterization are deficient and the results are poor and insufficiently substantiated.
In fact, as the main conclusion, the authors state that Co-MOF yields the best results - although this is only based on the specific capacitance values - but do not provide any explanation for this observation. Additionally, no chemical analysis is presented to confirm the presence of the different metallic cations in the structures.
Furthermore, SEM analysis suggests that only small clusters of MOF are deposited on carbon fibers, despite the various colors observed in Figure S1. If, indeed, this is the case, the electrochemical response of the coated materials would be predominantly influenced by the substrate, resulting in minor differences being observed.
In all MOF samples, the voltammograms exhibit a significant tilt concerning the XX axis, indicating a very high resistivity. This is further evident in the charge-discharge curves, which show a pronounced IR drop. The high materials resistivity suggests poor long-term performance of the materials although a comprehensive analysis in this regard was not conducted.
Finally, despite the authors' claim to be developing electrodes for structural supercapacitors, there is no supporting evidence, given the absence of mechanical testing or details regarding adhesion.
For these reasons, in its current form, I believe the manuscript does not meet the requirements for publication.
Thank you for your insightful comments and constructive feedback on our manuscript. The authors appreciate the opportunity to address the concerns raised during the review process. Here are our responses to each point, where an attempt has been made to improve the way in which the results are presented and to support the conclusions with the obtained results.
- “…Co-MOF yields the best results - although this is only based on the specific capacitance values…”
We sincerely appreciate your guidance and constructive feedback. Following your recommendations, we have thoroughly rewritten section “3.2 electrochemical properties of MOF-74 electrodes”. As suggested, we have not only tried to improve the overall English quality but also included the reasons behind our belief that Co-MOF-74 has exhibited best results compared to other metals.
- “…No chemical analysis is presented to confirm the presence of the different metallic cations in the structures…”
Your suggestion for including chemical analysis to confirm the presence of different metallic cations is valid. But, in this context, we believe that the performed characterization is sufficient to validate the successful synthesis of the desired material for several reasons. Firstly, the confirmation of correct synthesis of the MOF-74 structure is supported by the obtained X-ray diffraction pattern, which aligns with both simulated patterns and those reported by other researchers. Additionally, a secondary confirmation of the obtained structures comes from the measured BET surfaces areas, which again correlate with the values widely reported in the literature. Moreover, it is essential to note that the achievement of such elevated values in BET analysis would not be possible if the target structure had not been successfully synthesized.
Furthermore, the used MOF-74 structures inherently feature a single metal type, and it is crucial to emphasize that the absence of the incorporated metal in the material would prevent the desired structure from forming. So, after given the confirmation of the correct MOF-74 structure through two techniques (XRD and BET), coupled with the fact that only a single metal was added during the synthesis of each material, we deemed it unnecessary to conduct additional test to verify the proper incorporation of metals into the MOF-74 structures. Last but not least, each synthesis procedure for the different used materials was adapted from diverse scientific articles, all of which are duly referenced in Table 1.
- ” Furthermore, SEM Analysis suggests…”
We appreciate your insight into the SEM analysis and potential substrate influence. While it is true that Figure S1 displays coated carbon fiber fabrics (CFF) of varying dimensions, it is crucial to emphasize that our evaluation of their electrochemical performance has consistently involved immersing precisely the same CFF area in the electrolyte. This approach ensures a consistent and fair assessment of each electrode behavior.
Furthermore, we would like to highlight that the presented results in our manuscript are reported in terms of mass (F/g), taking into account the mass of each material. This consideration serves to minimize any potential distortion arising from variations in the substrate. By normalizing our properties based on mass, we aim to provide a more accurate comparison of the electrochemical properties of the MOF-coated carbon fiber electrodes
Regarding the deposition of MOF clusters on carbon fibers. While the authors agree with the reviewer that a more uniform coating could be considered a better result because it could lead to greater energy store capacity, our primary objective in this study was to identify the metal that exhibits superior electrochemical behavior (although we have not adequately expressed what this result is due to in the first version of the manuscript). Despite we also think that a continuous layer might be considered a more favorable outcome, in this work we presented our results taking into account the mass (F/g), and the main focus was to determine which metal yielded the best performance. In this context, the absence of a continuous layer was deemed a valid result due to the deposition of more mass would not modify the F/g ratio, so it did not impede the ability to draw meaningful conclusions related to our research question. Anyway, we value your suggestion regarding the pursuit of more homogeneous layers for future works, especially when our primary goal is to enhance the total energy storage capacity of the developed supercapacitor. We think that this suggestion will be duly considered in subsequent research endeavors, where different methods will be explored with the specific of increasing the overall energy storage capability of the supercapacitor.
We hope this clarification addresses your concerns, and we remain dedicated to ensuring the accuracy and reliability of our study
- “In all MOF samples, the voltammograms…“
Regarding the different MOF CV test, the authors agree with the reviewer comment. Due to the inherent degradation on the MOF-74 materials in water [40], it was impossible to employ water as an electrolyte. Consequently, we opted for a 2 M potassium chloride (KCl) solution in ethanol as the electrolyte medium. This electrolyte exhibits significant higher resistivity than if the same solution were prepared in water. Additionally, the amount of KCl could not be increased because the used concentration was close to the theoretical saturation value. Thus, the morphology of the CV curves is somewhat influenced by the used electrolyte. However, in any case, using the same electrolyte for all materials has allowed for a valid comparison between the synthesized materials.
On the other hand, concerning the charge and discharge test, the used electrolyte is even less conductive that the one mentioned earlier. However, the use of this electrolyte is justified because the aim is to prepare structural supercapacitor. Thus, the electrolyte consists of a blend of two epoxy systems (widely employed for structural purposes in CFRP) and an ionic liquid. So, the electrolyte is composed of 70 wt.% of a highly non-conductive epoxy resin (necessary for the structural behavior) with only 30 wt.% of ionic liquid, resulting in the low conductivity of the medium. However, this reduction in conductivity contributes to high mechanical resistance in the final composite material, resulting in the formation of a structural supercapacitor. So, not only a supercapacitor has been developed, a multifunctional material was also manufactured with mechanical and energy store capacity.
Finally, the authors also wish to emphasize that in this work the capacitance is associated with the total mass of the electrode, not solely the mass of MOF deposited on the electrode. This is why the resulting amperes are divided by a much larger mass value, yielding a lower capacitance value. Specifically, the mass of a 1 cm of a carbon fiber yard is 0.0020g, while the same fiber coated with Co-MOF-74 is 0.0023 g. Assuming a hypothetical example where we divide by the total electrode mass (0.0023 g) to calculate might obtain a result of 1 A/g. However, if we perform the same calculation considering only as the active mass the Co-MOF-74 (0.0003), capacitance will increase to 9.2 A/g. Because carbon fiber does exhibit some capacitance (Figure 4), albeit very low, the authors have chosen to present the data in a more conservative manner, including the mass of carbon fiber in capacitance calculations, even though the final result would be less numerically eye-catching.
- “Finally, …developing electrodes for structural supercapacitors … details regarding adhesion.”
The structural term has been employed because the used carbon fiber (Hexcel AS4C 3k 5H) and the epoxy systems (Ly556/XB3473 + PEDGE/DDS) are widely utilized in the fabrication on carbon fiber reinforced polymer (CFRP) composited in structural applications. Therefore, as a consequence of this, and since no damage has been observed on the CF surfaces in the SEM analysis due to the MOF deposition process (Figure 3), it can be concluded that the CFRP manufactured through VARIM possesses structural characteristics. The reviewer is correct in pointing out that mechanical test has not been reported in this work, but it is noteworthy that the assessment of the mechanical properties of these materials (reinforcement and matrix) was previously analyzed in our laboratory, as reported elsewhere [21]. Because of this fact, coupled with the absence of defects in the CF, it was considered that the obtained CFRP would retain the aforementioned structural characteristics. In any case, information regarding this discussion has been added to the new version of the manuscript (section 3.3), and if the reviewer deems it necessary, an analysis of the mechanical properties could be conducted.
Additionally, regarding the adhesion, the crystals observed on the CFF of this material exhibited a large size. This implies that, by not detaching from the fiber surface, they possessed sufficient anchoring forces to allow the subsequent manufacturing process of the SC. This is crucial because, thanks to this, the Co-MOF-74 crystals would not be carried away easily by the resin flow required to manufacture the supercapacitor through the vacuum assisted resin infusion molding (VARIM) process. So, if the material had been detached from the CF surface during the manufacturing process, the material carried by the resin flow would have been observed in the excess resin, and more importantly, storage energy capacity would not have been observed, and the proof-of-concept test could not have been conducted.
Reviewer 2 Report
Comments and Suggestions for Authors
In this submission, the authors report the fabrication of CFF supported MOF-74 and their applications in supercapacitors. Different active metal sites have been incorporated in MOF-74. The supercapacitors have been used as power sources for LEDs. The topic is interesting and could attract wide readership. Therefore, I recommend its publication after the following issues are addressed.
1. The BET surface area of MOF-74 is high, but the specific capacitance of the finally obtained supercapacitor is pretty low. The reasons should be analyzed in more detail.
2. The SEM images show that MOF-74 does not grow well on CFF. Therefore, the growth conditions could be improved for improving the growth of MOF-74.
3. CV curves of the MOF-74 supercapacitor at different scan rates show that there are significant polarization. This could be caused by the poor conductivity or large diffusion barrier.
4. The rate performance is pretty poor (Fig. 5b). The possible reasons should be discussed.
5. Why do Co-based MOF-74 have higher capacitance than other metal-based ones? Is it because of the morphology or the chemical composition?
6. The authors are recommended to cite relevant literatures such as Small 2022, 18, 2103866; J. Energy Storage 2023, 73, 109521.

Author Response
Thank you for your insightful comments and constructive feedback on our manuscript. The authors appreciate the opportunity to address the concerns raised during the review process. Here are our responses to each point.
- The BET surface area of MOF-74 is high, but the specific capacitance of the finally obtained supercapacitor is pretty low. The reasons should be analyzed in more detail.
Regarding the observed discrepancy between the high BET surface area of all the synthesized MOF-74 and the specific capacitance, the primary reason is associated with the low conductivity of the used electrolyte. The developed supercapacitor possesses a structural character, specifically composite material was manufactured (carbon fiber reinforced polymer). The carbon fiber electrodes act as the reinforcement for the composite material, while the matrix consists of a blend of two epoxy systems (widely employed for structural purposes) and an ionic liquid. Consequently, due the electrolyte is not only ionic liquid the conductivity of the medium, in this case, the matrix of the composite material, decreases dramatically. However, this reduction in conductivity contributes to high mechanical resistance in the final composite material, resulting in the formation of a structural supercapacitor. So, not only a supercapacitor has been developed, a multifunctional material was also manufactured with mechanical and energy store capacity.
On the other hand, in this work the capacitance is associated with the total mass of the electrode, not solely the mass of MOF deposited on the electrode. This is why the resulting amperes are divided by a much larger mass value, yielding a lower capacitance value. Specifically, the mass of a 1 cm of a carbon fiber yard is 0.0020g, while the same fiber coated with Co-MOF-74 is 0.0023 g. Assuming a hypothetical example where we divide by the total electrode mass (0.0023 g) to calculate might obtain a result of 1 A/g. However, if we perform the same calculation considering only as the active mass the Co-MOF-74 (0.0003), capacitance will increase to 9.2 A/g. Because carbon fiber does exhibit some capacitance (Figure 4), albeit very low, the authors have chosen to present the data in a more conservative manner, including the mass of carbon fiber in capacitance calculations, even though the final result would be less numerically eye-catching.
- The SEM images show that MOF-74 does not grow well on CFF. Therefore, the growth conditions could be improved for improving the growth of MOF-74.
The authors agree with the reviewer´s comment. After verifying the anchoring of the crystals on the CFF and having analyzed the effect of different metals (highlighting the favorable performance of cobalt, nickel and manganese), it is anticipated that future efforts will be directed towards modifying the synthesis to optimize the coating on CFF and thus enhance the supercapacitor performance.
- CV curves of the MOF-74 supercapacitor at different scan rates show that there are significant polarization. This could be caused by the poor conductivity or large diffusion barrier.
The authors appreciate the reviewer observation. Part of this question has been also previously detailed in question 1. The developed supercapacitor is a structural supercapacitor, and due to need to use of a blend of polymeric resins with the ionic liquid, the outcome leads to low conductivity, but achieving structural properties.
- The rate performance is pretty poor (Fig. 5b). The possible reasons should be discussed.
Similar to what was previously discussed, the main reasons for this behavior have been addressed in questions 1 and 3. Additionally, in Figure 5b, the effect of the scan rate is evaluated. In this case the drop in the result aligns with the expected behavior. Higher scan rates imply less time available for charging, resulting in lower values. Moreover, in calculating capacitance, it is necessary to divide by the mV/s value used in the measurement. Therefore, a higher scan rate leads to a lower result.
- Why do Co-based MOF-74 have higher capacitance than other metal-based ones? Is it because of the morphology or the chemical composition?
This behavior is a combination of various events. First, it is related to the fact that the Co-MOF-74 exhibited the best combination of properties in terms of specific surface area and pore volume. This allows for increased contact between the active material and the matrix, resulting in higher capacitance. Moreover, the use of cobalt-doped materials has also been previously emphasized by other authors due to the favorable enhancement in electrical conductivity provided by this metal, as referenced in the manuscript [48-52]. Additionally, the crystals observed on the CFF of this material exhibited a large size. This implies that, by not detaching from the fiber surface, they possessed strong anchoring forces. This is crucial because, thanks to this, the Co-MOF-74 crystals would not be carried away easly by the resin flow required to manufacture the supercapacitor through the vacuum assisted resin infusion molding (VARIM) process.
- The authors are recommended to cite relevant literatures such as Small 2022, 18, 2103866; J. Energy Storage 2023, 73, 109521.
Following the reviewer suggestions, after evaluating the suggested references, the proposed articles have been incorporated into the amended version of the manuscript.
Reviewer 3 Report
Comments and Suggestions for Authors
In this manuscript Authors report the synthesis of a series of MOFs functionalized with some transition metal cations on carbon fibres, in order to obtain electrode materials for structural supercapacitors.
Generally, the paper is well written, clear and exhaustive. The syntheses are well described, and the materials’ characterizations are appropriate. Figures are both eye-catching and explicative; conclusions are sound. For these reasons, I think the manuscript should be considered for publication in Nanomaterials. However, in my opinion a few minor revisions are necessary.
1. Figure 1 could benefit from a higher resolution.
2. Paragraph Electrochemical properties of MOF-74 electrodes should be 3.2 and not again 3.1.
3. This same Electrochemical properties of MOF-74 electrodes paragraph somehow seems of a lower quality in terms of English language; its reading is much less fluid as the sentence construction is less linear. Please try to rephrase some of the concepts, to align this section with the previous ones.
Comments on the Quality of English LanguageThe Electrochemical properties of MOF-74 electrodes paragraph somehow seems of a lower quality in terms of English language; its reading is much less fluid as the sentence construction is less linear. Please try to rephrase some of the concepts, to align this section with the previous ones.
Author Response
Thank you for your insightful comments and constructive feedback on our manuscript. The authors appreciate the opportunity to address the concerns raised during the review process. Here are our responses to each point.
- Figure 1 could benefit from a higher resolution.
Following the reviewer suggestions, Figure 1 has been modified.
- Paragraph Electrochemical properties of MOF-74 electrodes should be 3.2 and not again 3.1.
The authors appreciate the reviewer comment and have addressed the mentioned error in the manuscript.
- This same Electrochemical properties of MOF-74 electrodesparagraph somehow seems of a lower quality in terms of English language; its reading is much less fluid as the sentence construction is less linear. Please try to rephrase some of the concepts, to align this section with the previous ones.
Following the reviewer suggestion, the section 3.2 has been rewritten to make the text easier to read, eliminating complex and repetitive phrases.
Round 2
Reviewer 1 Report
Comments and Suggestions for Authors
Although I still think that the present work could be more comprehensive, specifically by evaluating the long-term behavior of the supercapacitor (which I believe would be very poor), I consider that the authors have justified all my other comments. In this view, I think that the work can be published in Nanomaterials."
Reviewer 2 Report
Comments and Suggestions for Authors
The authors have addressed the issues raised in the comments. Therefore, I recommend its publication without further revision.
Comments on the Quality of English LanguageThe language is OK for this manuscript.